# Revisiting the Impact of Corporate Carbon Management Strategies on Corporate Financial Performance: A Systematic Literature Review

**Maruli Sitompul** [1] , **Arif Imam Suroso** [1,*] , **Ujang Sumarwan** [2] **and Nimmi Zulbainarni** [1]

[1] School of Business, IPB University, Bogor 16151, Indonesia; marulisitompul@apps.ipb.ac.id (M.S.)
[2] Faculty of Human Ecology, IPB University, Bogor 16680, Indonesia
[*] Correspondence: arifimamsuroso@apps.ipb.ac.id

**Abstract:** The objective of this research is to examine the relationship between carbon management strategies in corporations and their impact on financial performance. We employ a systematic literature review to analyze 223 articles retrieved from reputable journals indexed in Scopus. A total of 22 empirical studies covering various industry sectors and countries were selected and included in our analysis. The result indicates that 59% of the articles demonstrate positive findings. Among these, 50% show a significant positive impact, while 9% exhibit mixed results with both positive and negative outcomes in the short and long-term perspectives. These findings suggest that adopting carbon management strategies predominantly has a positive influence on corporate financial performance. In this study, we also provide a summary of the dependent, independent, and control variables, as well as commonly used indicators in this research topic, to help guide future quantitative research. Lastly, we offer a summary of the motivations, drivers, and barriers that corporations experience when implementing carbon management strategies. These insights will be valuable for business managers and policymakers, aiding them in successfully embarking on the journey to achieve net-zero emissions.

**Keywords:** GHG emissions; climate change strategies; corporate carbon strategies; carbon management; carbon performance; financial performance

## 1. Introduction

Climate change is occurring at a much faster rate, leading to increased severity of natural disasters and extreme weather events (IPCC 2023; Climate Impact Partners 2023). This puts businesses at greater risk and threatens their operations, assets, and the well-being of their people. The Intergovernmental Panel on Climate Change (IPCC 2021) unequivocally asserts that anthropogenic emissions are contributing to the escalating intensity of climate change. This situation leads to a growing urgency for businesses to embark on and take immediate climate action.

In December 2015, countries united under the United Nations (UN) committed to the legally binding Paris Agreement, a global treaty addressing climate change. The Paris Agreement aims to achieve net zero emissions by 2050, limiting global temperature rise to well below 2 °C and striving to limit it to 1.5 °C (UNFCCC 2015). In response to this commitment, the past decade has witnessed an accelerating global effort to achieve the net-zero target by 2050. Currently, 195 countries have established and declared their national emission reduction targets, known as Nationally Determined Contributions (NDCs), in alignment with the Paris Agreement's objectives (https://unfccc.int/NDCREG (accessed on 2 May 2023)).

In addition to national efforts, global corporations have displayed their commitment and ambitious targets to achieve net zero emissions by 2050. A report highlights that 100 global companies contributed to 71% of global greenhouse gas (GHG) emissions

between 1998 and 2015 (CDP 2017). Mounting pressure from stakeholders such as governments, regulations, investors, financial institutions, media, public opinions, customers, suppliers, and employees has compelled companies to incorporate GHG emissions reduction into their management strategies (Buysse and Verbeke 2003; Cadez et al. 2018). This strategic approach is commonly referred to as a carbon management strategy in the literature (Yunus et al. 2020; Dhanda and Malik 2020). Various terms have been used to describe these strategies, including climate change strategy (Hoffman 2005; Michalisin and Stinchfield 2010; Backman et al. 2017), corporate $CO_2$ strategy (Weinhofer and Hoffmann 2010), corporate carbon strategies (Lee 2012; Damert et al. 2017), low-carbon operations management (Böttcher and Müller 2015; Alves et al. 2017), green management (Gramkow and Anger-Kraavi 2019), and decarbonization (de Sousa Jabbour et al. 2018; Wójcik-Jurkiewicz et al. 2021).

An increasing number of companies have adopted carbon management strategies and integrated them into their core business operations. They have observed that investments made in implementing carbon management strategies yield benefits such as cost efficiency, higher productivity, product differentiation, and increased revenues (Ambec and Lanoie 2008; Weinhofer and Hoffmann 2010), providing a competitive advantage for the company (Schultz and Williamson 2005). However, many companies continue to conduct business as usual and have not yet transitioned to carbon reduction commitments. The main reason is identified as the pervasiveness of short-term profit maximization (Damert et al. 2017). This fundamental business objective conflicts with the long-term goal of reducing carbon concentration in the atmosphere to mitigate climate change. From the perspective of corporations, implementing carbon reduction strategies requires substantial investments that could potentially erode the company's profitability (Slawinski et al. 2017).

To date, there has been relatively little effort to systematically clarify the impact of carbon management strategies on a company's financial performance. The central question that both researchers and business practitioners pose is whether it is financially beneficial to be green or environmentally responsible (Orsato 2006; Ambec and Lanoie 2008). Does the implementation of carbon management strategies have a positive impact on a company's financial performance? In an era of increasing demand for accelerated corporate climate action, understanding this relationship has become highly relevant and crucial. This research aims to investigate this relationship and provide a framework and an overview of the motivations, drivers, and barriers for researchers and business practitioners when adopting carbon management strategies. Our study reveals that there are still relatively few systematic literature reviews (SLRs) examining the empirical relationship between corporate carbon management strategies and financial performance. This paper represents the first analysis of its kind and is highly relevant in the current climate action landscape. We believe that this study will enrich the existing theory on the relationship between carbon management strategies and corporate financial performance, offering insights and understanding that researchers and corporate practitioners have been seeking.

To structure the objective of this research article, we have formulated several key research questions (RQs) that will be investigated in this study:

RQ1. What are the prevailing theories utilized to support research on corporate carbon management strategies and their impact on corporate financial performance?

RQ2. What do the findings from the literature reveal concerning the relationship between carbon management strategies and financial performance from 2016 to 2022 after the Paris Agreement 2015?

RQ3. Which variables and indicators are employed in quantitative studies focusing on carbon management strategies in corporate settings?

RQ4. What are the motivations, drivers, and barriers influencing the adoption and outcomes of corporate carbon strategies?

RQ5. (a) What potential opportunities exist for future research that could contribute to advancing the academic discourse on corporate carbon management? (b) What are the limitations inherent in the existing literature?

Based on the aforementioned research questions, this study aims to contribute to the advancement of the current literature regarding corporate carbon strategies and their impact on the financial performance of companies. Additionally, this research will provide insights into the motivations, drivers, and barriers that companies encounter during the adoption and implementation of carbon reduction strategies and the specific strategies and practices employed by these companies. The understanding gained from this study holds significant importance and applicability for practitioners and managers in corporate settings, enabling them to effectively prepare for the transition towards low-carbon business operations and the attainment of long-term targets for achieving net zero emissions goals.

We organized this article as follows. It begins with an explanation of our research methodology, following the Preferred Reporting Items for Systematic Review and Meta-Analysis (PRISMA) guidelines for SLR (Section 2). The results and discussion of the study are presented in Section 3, including answers to our research questions. Section 4 concludes the paper, outlining future research opportunities and acknowledging the limitations of our research.

## 2. Methodology

The study employs PRISMA guidelines, which facilitate a structured and systematic approach to the literature review. This ensures transparency and the utilization of evidence-based practices throughout the process of identifying, selecting, appraising, and synthesizing the reviewed studies (Kayan-Fadlelmula et al. 2022). The PRISMA method was initially developed in 2005 and has primarily been applied in medical articles which adhere to four key systematic steps: identification, screening, eligibility, and inclusion (Moher et al. 2009). These steps are commonly depicted using a flow diagram that illustrates the sequential application of inclusion and exclusion criteria to the collected articles (Moher et al. 2009). While PRISMA is well-established in medicine and healthcare, its utilization has gained popularity in recent years across various disciplines, including social sciences, management, environmental studies, and other fields. According to Ismail et al. (2021), there are three advantages to employing PRISMA in SLR studies: (1) it facilitates the formulation of focused and clear research questions, (2) it enables the systematic screening of articles based on predefined inclusion and exclusion criteria, and (3) it enables the selection of relevant literature from diverse scientific journal databases within a specified timeframe.

The search for papers in this study utilizes the Scopus database. We restrict ourselves to the Scopus database because, after reviewing several research articles, it has been demonstrated by some authors that Scopus is more suitable for bibliometric analysis compared to other databases, i.e., the Web-of-Science (WoS) database (Gao et al. 2021). This preference is due to several reasons: Scopus maintains high-quality standards, provides broad coverage of information, and offers easy data downloading capabilities (Herrera-Franco et al. 2020). Additionally, Scopus stands out for its extensive citation database, encompassing over 20,000 peer-reviewed journals (Wahyudi et al. 2021). Mishra et al. (2017) also affirm that the Scopus database is more comprehensive compared to WoS, especially in the field of green, sustainability, and the environment. The research articles included in this collection cover the period from 2016 to 2022. The rationale for selecting this review period was to examine the impact of the 2015 Paris Agreement, a globally binding commitment signed in Paris, which mandates all countries to strive to achieve net-zero emissions by 2050. This agreement also encompasses the commitments made by corporations in relation to climate change actions.

The search for the papers included in this article was conducted on 7 September 2022. Our SLR study adheres to the four steps outlined in PRISMA (Liberati et al. 2009):

- Identification: searching for all relevant articles from selected or various databases.
- Screening: selecting articles based on a defined set of criteria to include or exclude.
- Eligibility: ensure that all screened articles conform to the eligibility criteria.
- Inclusion and exclusion: making the final selection of the eligible articles for analysis.

## 2.1. Identification

The first step in the identification process is to define the keywords. These keywords are selected to accurately represent the scope and breadth of the topic we aim to address. In this study, we generated search strings using the chosen keywords in the Scopus database (www.scopus.com (accessed on 7 September 2022)). The selected keywords include GHG Emissions, Climate Change Strategies, Corporate Carbon Strategies, Carbon Performance, and Financial Performance. The search formula used: TITLE-ABS-KEY ("ghg emission*" OR "climate change strateg*" OR "corporate carbon strateg*" OR "carbon management" OR "carbon performance" AND "performanc*" AND "financial"). Utilizing these keywords, we identified a total of 223 articles from Scopus.

## 2.2. Screening

The second step involves the screening phase, during which a set of criteria is established to determine the inclusion or exclusion of the identified articles. These criteria encompass the publication year, document type, publication stage, country of origin, source type, and language. In this phase, the decision was made to exclusively analyze articles while excluding books and book reviews to ensure the research's quality and validity. Academic articles undergo a rigorous peer-review process, employ scientific methodologies, and are grounded in empirical studies, thus providing reliable and up-to-date information. Furthermore, they offer more precise and targeted insights that align with the objectives of our SLR. The comprehensive results of the inclusion and exclusion process, based on the defined criteria, are presented in Table 1. As a result of this process, from total of 223 articles screened, the researchers included 105 articles while excluding 118 articles.

**Table 1.** The criteria for inclusion and exclusion in the screening step.

| Criteria | Inclusion | Exclusion |
|---|---|---|
| Year published | Before 2016 and beyond September 2022 | 2015 and before |
| Document type | Article | Book Chapter, Conference Paper, Abstract Report, Review, Letter, Conference Review. |
| Publication stage | Final | Article in Press |
| Country/territory | All countries | |
| Source type | Journal | Proceeding of Conference, Book, Trade Journal, Book Series. |
| Language | English | Non-English |

## 2.3. Eligibility

Based on the results of the screening phase, we progressed to the third step, known as the eligibility step. In this step, a comprehensive evaluation of the 105 articles was conducted. The examination involved reviewing the titles, abstracts, methods, and results of each selected article to ascertain their alignment with the predefined inclusion criteria. During this process, our focus was on selecting articles that directly addressed corporate strategy, in line with the objectives of our current research. As a result, a total of 57 articles were excluded based on this criterion. Among the excluded articles, 26 focused on the scope of a city, country, or region in terms of policy and implementation, 29 discussed specific technologies and processes, 1 article lacked the variables related to carbon strategies and their impact on financial performance, and 1 article was either an LSR or a meta-analysis. After this eligibility process, 48 articles remained.

## 2.4. Inclusion

In the final step, we conducted a thorough assessment of the full-text articles from the initial selection of 48 articles. This assessment involved reading both the abstracts and the complete contents of each article. We examined whether the full-text articles presented

empirical research, employed quantitative methods, discussed relevant variables and indicators, and reported the results pertaining to the relationship between corporate carbon strategies and financial performance. As a result of this evaluation, we excluded 26 articles that did not meet these criteria. Consequently, the total number of articles included in this SLR study is 22.

### 3. Results

For the SLR, we obtained a total of 22 selected articles spanning the years from 2016 to 2022. The selection of articles for inclusion in our study was determined through two key stages of the PRISMA methodology. The first stage is the eligibility stage, during which we applied the eligibility criteria to the articles identified in the screening stage. A total of 105 articles were evaluated against the eligibility criteria, resulting in 48 articles that met the criteria and progressed to the next stage. The second stage is the inclusion stage, in which we further refined the selection by applying the inclusion criteria. From the 48 articles, 26 were excluded based on the inclusion criteria, leaving us with the final set of 22 articles for our study. The complete process of article selection, following the PRISMA method, is illustrated in Figure 1.

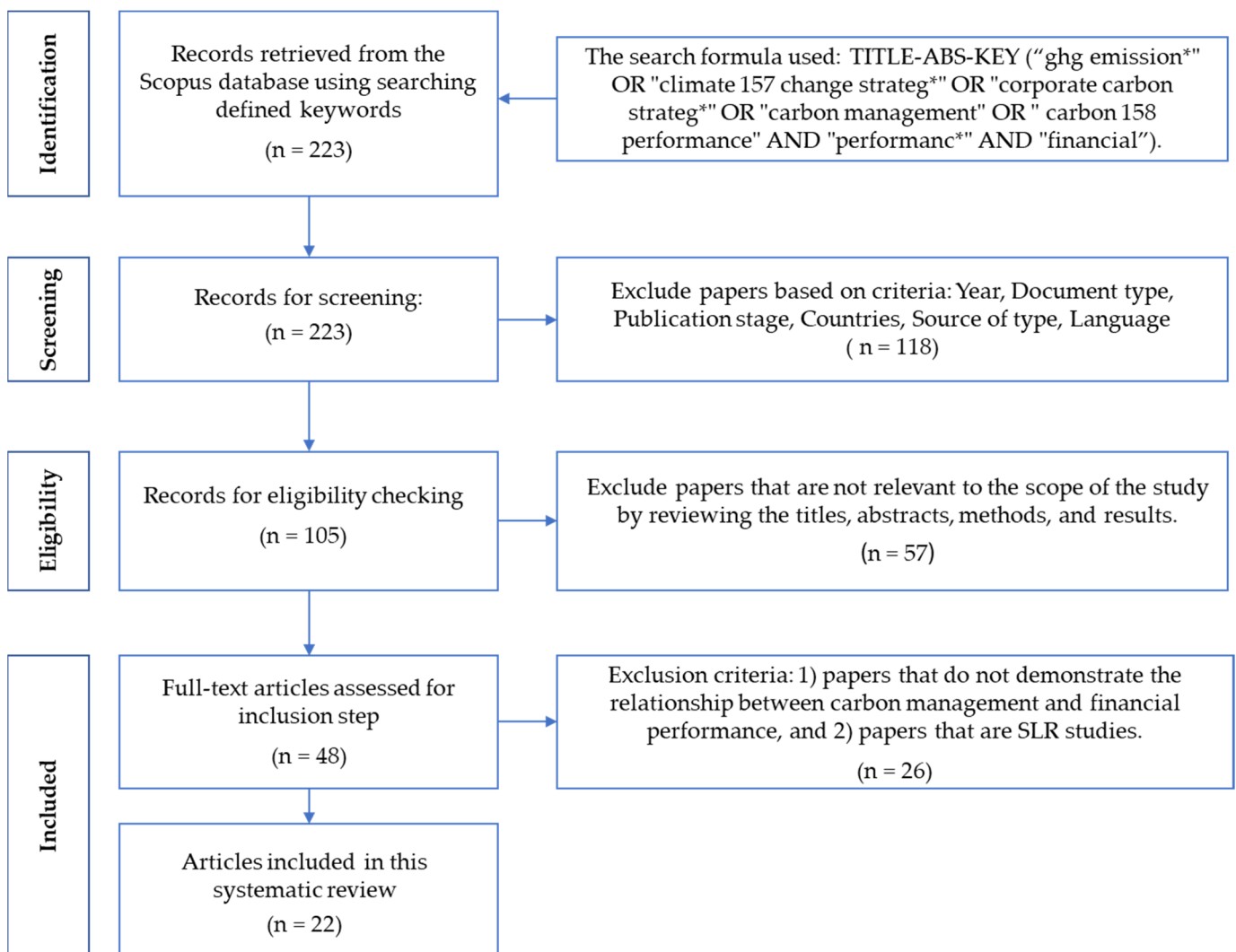

**Figure 1.** Flow chart of the study selection process using PRISMA.

Tracking the annual publication of articles (Figure 2), we observed a consistent trend of studies focusing on the relationship between corporate carbon management and corporate

performance. The year 2017 had the highest number of published articles (7). Although the number of articles decreased in the subsequent years from 2018 to 2022, there was still a consistent publication of several articles, averaging three articles per year.

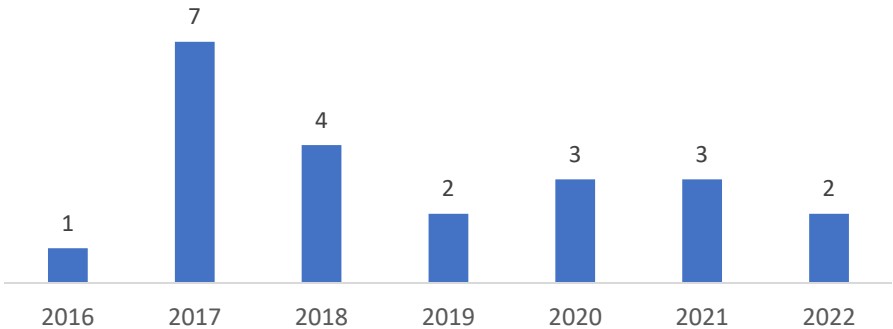

**Figure 2.** Distribution of articles by years.

All selected articles were published in high-rated journals indexed by Scopus, with more than 95% of them originating from journals ranked in Q1 and Q2 (Table 2). The articles were published in the following journals: Business Strategy and the Environment (7), Journal of Cleaner Production (4), Environmental Development (1), Environmental Science and Pollution Research (1), International Review of Financial Analysis (1), Organization and Environment (1), Social Responsibility Journal (1), Sustainability (1), International Journal of Energy Economics and Policy (2), Australasian Journal of Environmental Management (1), China Journal of Accounting Studies (1).

**Table 2.** The source of journals where articles were selected.

| Source of Journal | Number of Articles | Quartiles | SJR 2021 |
|---|---|---|---|
| Business Strategy and the Environment | 7 | Q1 | 2.24 |
| Journal of Cleaner Production | 4 | Q1 | 1.92 |
| Environmental Development | 1 | Q1 | 0.91 |
| Environmental Science and Pollution Research | 1 | Q1 | 0.83 |
| International Review of Financial Analysis | 1 | Q1 | 0.83 |
| Organization and Environment | 1 | Q1 | 1.62 |
| Social Responsibility Journal | 1 | Q1 | 0.63 |
| Sustainability (Switzerland) | 1 | Q1 | 0.66 |
| International Journal of Energy Economics and Policy | 2 | Q2 | 0.38 |
| Australasian Journal of Environmental Management | 1 | Q2 | 0.44 |
| China Journal of Accounting Studies | 1 | Q3 | 0.28 |
| Grand Total | 22 | | |

When examining the country origins (Table 3), it is evident that the studies on corporate carbon management primarily originate from research conducted in European countries, while other regions of the world have fewer studies. This indicates that the most advanced studies and implementation of corporate climate strategies for carbon reduction are predominantly found in this region. Among the observed countries, there are 12 articles related to the European region. Among them, five articles explored various selected countries within the European region, four articles focus on European Union (EU) countries exclusively, two articles concentrate on the United Kingdom (UK), and one article focuses on Italy. The European region has implemented the majority of strategies and policies pertaining to climate change mitigation and carbon reductions. The Emission Trading System (ETS) and carbon trading in the European Union serve as pioneering mechanisms that have become benchmarks for other regions. Additionally, some countries have begun imposing carbon fines on emissions generated by companies (Tahat and Mardini 2021). In the UK,

regulations have been put in place to make carbon disclosures mandatory for companies, ensuring transparency in their commitment to carbon reduction (Gerged et al. 2020).

**Table 3.** The distribution of selected articles by country of origin.

| Countries Studied | Frequency |
|---|---|
| Multi-countries | 5 |
| European Union countries | 4 |
| Indonesia | 3 |
| Australia | 2 |
| South Africa | 2 |
| United Kingdom | 2 |
| Canada | 1 |
| Italy | 1 |
| Japan | 1 |
| USA | 1 |

Similar mandatory carbon reductions have been mandated in select EU member countries, Australia, and Japan (Siddique et al. 2021), with their respective annual reports providing additional information on the progress made in carbon reduction. In contrast, in other countries, the implementation of these reductions by most companies remains voluntary. The significance of carbon disclosure has increased in recent years, as it aids companies in effectively communicating their commitment to mitigating climate change, as expected by stakeholders, investors, and financial institutions (Borghei et al. 2018; Alsaifi et al. 2019).

When examining the industry sectors studied (Table 4), it was found that the majority of articles (73%) focused on investigating the relationship between corporate carbon management strategies and corporate performance in a multi-industry context. This indicates that the findings are applicable across a wide range of industries. This multi-industry category includes various sectors such as oil and gas, basic materials, utilities, steel and cement, automotive, consumer goods, health care, telecommunication, computer hardware, computer services, financials, and consumer services. By including multiple industries in their samples, these articles provide more robust and generalizable conclusions. The remaining articles, accounting for 18% of the total, exclusively examined the manufacturing industry, while 4.5% of the articles specifically focused on the food, dairy, and automotive industries.

**Table 4.** The distribution of the articles by industry sectors.

| Industry Sector | Frequency | Percentage |
|---|---|---|
| Automotive | 1 | 4.5% |
| Dairy farming | 1 | 4.5% |
| Manufacturing | 4 | 18% |
| Multi-industries | 16 | 73% |

## 4. Discussion

In this section, our objective is to present and analyze the primary findings of the study, addressing the research questions outlined in Section 1 (Introduction). The questions that we aim to explore and discuss are as follows:

- The utilization of theories in all selected research articles.
- The result obtained regarding the relationship between carbon management strategies and financial performance.
- The shared variables and indicators employed in the selected studies.
- The motivations, drivers, and barriers encountered by companies when implementing corporate carbon strategies.

To facilitate the discussion, we will address each research question individually, providing comprehensive answers and insights based on our findings.

### 4.1. The Theories in the Literature on Carbon Management Strategy and Corporate Performance

This study examined the supporting theories commonly utilized in research pertaining to carbon management strategies within corporates. Most of the articles employed a single theory, while some incorporated combined theories (multiple theories). However, a few articles lacked clarity in discussing and mentioning the theory employed.

In their research, Damert et al. (2017) mentioned the utilization of multiple theories, including stakeholder theory, institutional theory, legitimacy theory, and (natural) resource-based theory. The stakeholder and institutional theories elucidate the significant pressures exerted by stakeholders and institutions, compelling companies to adopt and comply with carbon management strategies. By demonstrating and communicating their commitment to addressing climate change issues, companies can differentiate themselves from competitors and potentially reap long-term financial benefits (Kolk and Pinkse 2007; Busch et al. 2022).

The final theory is the utilization of the (natural) resource-based view theory, which proposes that firms leverage all their resources, investments, differentiation, and unique qualities to attain competitive advantages. In Ganda's (2018) study, the impact of carbon performance on corporate financial performance is examined. The study emphasizes the use of multiple interconnected theories, including legitimacy theory, stakeholder theory, and institutional perspectives, to comprehensively understand the influence of environmental factors on a company. Through the integration of these theories, a framework is established to differentiate between macro and micro-factors that can potentially affect carbon performance initiatives within a corporate context.

Table 5 provides a summary of theories commonly employed by researchers in their studies to explain carbon management strategies within corporations. The analysis reveals five main theories frequently utilized in this research domain: stakeholder theory (Damert et al. 2017; Rokhmawati and Gunardi 2017; Lewandowski 2017; Trumpp and Guenther 2017; Ganda 2018; Siddique et al. 2021; Tuesta et al. 2021), institutional theory (Damert et al. 2017; Damert and Baumgartner 2018; Ganda 2018; Tuesta et al. 2021), natural resource-based view theory (He et al. 2016; Lewandowski 2017; Trumpp and Guenther 2017; Tuesta et al. 2021) and resource-based view theory (Borghei et al. 2018; Alsaifi et al. 2019) and legitimation theory (He et al. 2016; Ganda 2018; Tuesta et al. 2021; Siddique et al. 2021) are identified as prevalent theories. Additionally, several other theories are observed in the selected articles, including agency theory (Fernández-Cuesta et al. 2019; Tuesta et al. 2021), pecking order theory (Fernández-Cuesta et al. 2019), trade-off theory (Fernández-Cuesta et al. 2019), signal theory (He et al. 2016; Siddique et al. 2021), voluntary disclosure theory (Siddique et al. 2021; Tahat and Mardini 2021), and competitive advantage theory (Rokhmawati and Gunardi 2017).

**Table 5.** Grand theories used in carbon management research.

| Grand Theories | Authors |
| --- | --- |
| Agency theory | Fernández-Cuesta et al. (2019); Tuesta et al. (2021) |
| Pecking order theory | Fernández-Cuesta et al. (2019) |
| Trade-off theory | Fernández-Cuesta et al. (2019) |
| Institutional theory | Damert et al. (2017); Damert and Baumgartner (2018); Ganda (2018); Tuesta et al. (2021) |
| Stakeholder theory | Damert et al. (2017); Rokhmawati and Gunardi (2017); Lewandowski (2017); Trumpp and Guenther (2017); Ganda (2018); Siddique et al. (2021); Tuesta et al. (2021) |
| Natural resource-based view theory | He et al. (2016); Lewandowski (2017); Trumpp and Guenther (2017); Tuesta et al. (2021) |
| Resource-based view theory | Borghei et al. (2018); Alsaifi et al. (2019) |
| Legitimacy theory | He et al. (2016); Ganda (2018); Tuesta et al. (2021); Siddique et al. (2021) |
| Signal theory | He et al. (2016); Siddique et al. (2021); Damert et al. (2017) |
| Voluntary disclosure theory | Siddique et al. (2021); Tahat and Mardini (2021) |
| Competitive advantage theory | Rokhmawati and Gunardi (2017) |

### 4.2. The Relationship between Corporate Carbon Management and Financial Performance

Upon examining the selected articles, it becomes evident that the results of studies investigating the relationship between corporate carbon management and corporate performance vary. Figure 3 illustrates the distribution of findings across the studies. Among the articles, 50% indicate a positive relationship (11 articles), 27% reveal a negative relationship (6 articles), 5% do not provide a clear observation (1 article), 9% exhibit a dual result characterized by a positive relationship in the long-term but a negative relationship in the short-term (2 articles), and finally, 9% exhibit a non-linear U-shaped relationship (2 articles).

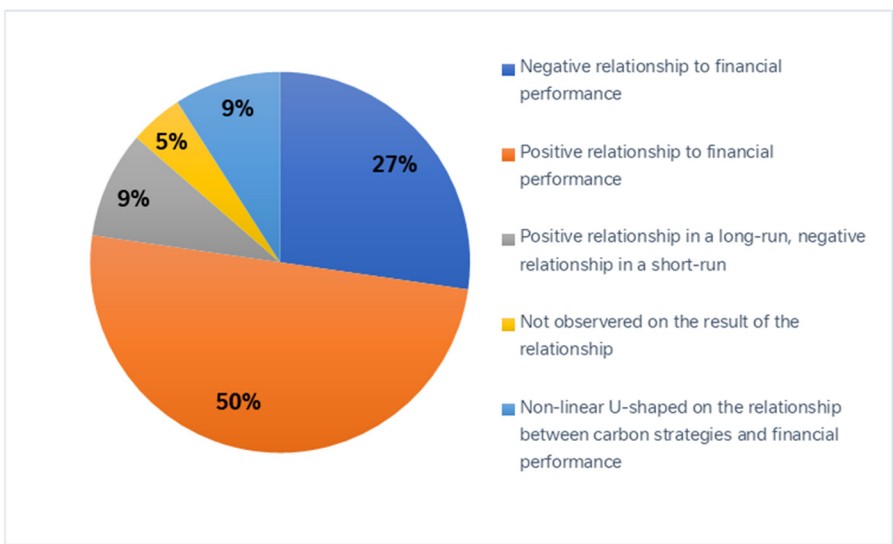

**Figure 3.** Distribution of result study on the relationship between carbon performance and financial performance (in percentage).

Hoffman (2005) conducted a study on GHG reduction initiatives implemented in the operations of various global corporations across different industries. The study emphasized the effectiveness of initiatives aimed at reducing energy use, such as the installation of energy-efficient heating and lighting systems, as well as the implementation of energy efficiency programs. These initiatives not only contribute to GHG emissions reduction but also result in significant cost savings. For example, in the United States, where lighting accounts for up to twenty percent of electricity consumption (Hoffman 2005), these examples provide valuable insights into the efficiency opportunities that companies can explore to achieve similar benefits.

The study conducted by Rokhmawati et al. (2017) showed a significant positive effect of $CO_2$e intensity on the return on sales (ROS). This finding suggests that customers respond positively to companies' efforts to reduce their greenhouse gas (GHG) emissions.

Similarly, He et al. (2016) find a positive and significant relationship between carbon reduction and financial performance. Their study demonstrates that the implementation of green projects tends to simultaneously mitigate emissions and generate observable financial benefits, as reflected in an improved Tobin's Q. These results indicate that stakeholders and the market communicate their concern for global warming issues and recognize proactive climate change strategies as key factors for survival and achieving sustainable economic success.

In addition to the positive findings, it is important to consider the results in terms of the time perspective. For instance, Ganda (2022) reveals that in the short run, carbon performance exhibits a significantly positive link with return on assets (ROA), firm value, and Tobin's Q. However, in the long run, the relationship between carbon performance and ROA, as well as firm value, becomes negative. Typically, short-term financial performance is measured using the indicator of ROA, while Tobin's Q is utilized for assessing

long-term financial performance (He et al. 2016; Lewandowski 2017; Siddique et al. 2021; Busch et al. 2022; Ganda 2022).

In contrast, Tuesta et al. (2020) present results indicating a negative and significant linear correlation between carbon management variables and profitability variables, particularly in non-sensitive industries. Moreover, Busch et al. (2022) find that higher carbon emissions are associated with increased short-term and long-term financial performance. However, Damert et al. (2017) find no relationship between the implementation of carbon management strategies and long-term improvements in corporate carbon performance. Additionally, these two indicators are not linked to long-term financial gains for companies.

*4.3. The Variables and Indicators Used in Quantitative Research Literature Studied*

From the selected 22 articles published during 2016–2022, we summarized the common variables and their indicators used for research studies on corporate carbon management and corporate performance (Table 6). In the table, we summarize all variables used, both independent and dependent, and their assigned indicators. We also included the control variables that are important but are controlled because they could influence the outcomes.

In the table list, we can see that the empirical studies are taken using various ranges of years samples, with an average of 4.5 years, ranging from 1 year to 12 years. In the majority, the articles study more than 100 sample companies.

Finally, we provided the correlation result of each selected article on the relationship between corporate carbon strategies and corporate financial performance. Common independent variables used in the research are corporate carbon performance or carbon performance, environmental performance, carbon management measures, Corporate Environmental Performance (CEP), GHG emissions, carbon footprint, emissions reduction, GHG disclosure, and carbon disclosure. The common dependent variables used that are linked to corporate performance are financial performance, Tobin's Q, financial debt, Corporate Financial Performance (CFP), and economic performance. At the same time, the control variables that are commonly used are firm age, firm size, leverage, growth, type of firm, type of industry, domestic market orientation, capital intensity, carbon intensity, board size, and liquidity.

**Table 6.** Summary of variables and indicators used in the selected articles and their results.

| Authors | Independent | | Dependent | | Controls | Correlations | Year Sample | Number of Samples |
| | Variables | Indicators | Variables | Indicators | | | | |
| --- | --- | --- | --- | --- | --- | --- | --- | --- |
| (Busch et al. 2022) | Corporate carbon performance | Total GHG Emissions for direct and indirect scopes | Financial performance | ROA | Growth, size, leverage, capital insensitive. | Negative | 2005–2014 | 27,986 firm-year, 4873 companies |
| (Ganda 2022) | Carbon performance | Carbon performance | Financial performance | Return on assets, firm values, Tobin's Q | Debt-to-equity ratio, interest cover, price-to-cash flow, and current ratio | Positive in a long run, negative in a short run | 2014–2018 | 107 firms |
| (Siddique et al. 2021) | Carbon performance | Carbon disclosure scores; Carbon performance | Financial performance | ROA, Tobin's Q | Firm age, firm size, capital intensity, leverage, earnings quality, Stock liquidity and carbon intensity | Positive in a long run, negative in a short-run | 2011–2015 | 187 firms |
| (Tuesta et al. 2021) | Total emissions, the ratio of emissions, direct emissions, indirect emissions, environmental certificate | Tons of $CO_2$ reported, emission/sales, tons of direct emissions, tons of indirect emissions | Tobin's Q | (Market value + liquidation value share + current liabilities + long-term debt)/total assets | Growth, size, leverage, capital intensities, age | Negative | 2007–2018 | 350 firms |
| (Tahat and Mardini 2021) | Carbon performance | Corporate carbon disclosure score, carbon emissions (Scope 1, 2, 3) | Financial performance | ROA, ROE | Size, liquidity, leverage, growth, and dividends per share | Positive | 2015–2018 | 116 firms |
| (Alsaifi et al. 2019) | Carbon disclosure | | Financial Performance | ROA, ROE, asset turnover, debt to equity ratio, interest coverage, return volatility, cost of equity, price earnings ratio, market to book ratio. | Firm size, leverage, market competition, foreign market activities, board size | Positive | 2007–2015 | 977 firm years |

**Table 6.** *Cont.*

| Authors | Independent | | Dependent | | Controls | Correlations | Year Sample | Number of Samples |
|---|---|---|---|---|---|---|---|---|
| | Variables | Indicators | Variables | Indicators | | | | |
| (Qian et al. 2020) | Environmental performance | Carbon Performance, Industry Sensitivity (IndSen), Environment Sensitivity (NGER) | Financial performance | Cumulative Abnormal Stock Returns (CARs) | Market capitalization, market to book value, growth rate, | Positive | 2009–2010 2011 2013–2014 | 1475 firms for the ETS event 1672 firms for carbon tax events 1842 firms for repeal of carbon tax events |
| (Tuesta et al. 2020) | Carbon management measures | Score emissions, emissions, variation of emissions, carbon performance | Financial performance | ROA, ROE, ROS | Global Reporting Initiative, variation in property, plant, and equipment, growth, market share, leverage, size, operational expenses, type of legislation. | Negative | 2006–2017 | 497 firms |
| (Fernández-Cuesta et al. 2019) | Environmental performance | Carbon performance | Financial debt | Sales, profitability, investment intensity | Profitable, size, tangible assets, non-debt tax shields, R&D expenses, firm age, liquidity, and the corporate tax rate in each country | Negative | 2005–2012 | 4223 firm year, 428 firms |
| (Jayasundara et al. 2019) | Carbon footprint | GHG emissions | Financial performance | Profit | None | Negative | 2010–2012 | 182 farms |
| (Damert and Baumgartner 2018) | Emissions reduction | Governance, innovation, compensation, legitimation | Financial performance | ROA, ROE | Firm size, financial performance | Not observed | 2013–2014 | 117 firms |
| (Yagi and Managi 2018) | Corporate Environmental Performance (CEP) | Carbon intensity, energy intensity | Corporate Financial Performance (CFP) | ROA | TATR, leverage, firm size | Positive | 2011–2015 | 225 firms |

**Table 6.** *Cont.*

| Authors | Independent | | Dependent | | Controls | Correlations | Year Sample | Number of Samples |
|---|---|---|---|---|---|---|---|---|
| | Variables | Indicators | Variables | Indicators | | | | |
| (Borghei et al. 2018) | GHG Disclosure | GHG Disclosure Score | Financial performance | ROA, ROE, ROS | Age, firm size, leverage, enterprise value, and capital intensity. | Positive | 2009–2011 | 290 firms |
| (Ganda 2018) | Carbon performance | CP Rating | Financial Performance | ROE, ROI and ROS, Market Valuation | Firm size, capital intensity, leverage, growth | Positive | 2014–2015 | 63 firms |
| (Trumpp and Guenther 2017) | CEP | Carbon performance, waste intensity | CFP | TSR, ROA | R&D intensity, capital intensity, leverage, growth, cash flow, company size, legal origin | Non-linear U-shaped relationship | 2008–2012 | 1179 firm years |
| (Lewandowski 2017) | Carbon performance | Carbon performance ($CO_2$ intensity) | Financial performance | Profitability (ROA, ROE, ROS, ROIC), market performance (Tobin's Q) | Firm size, risk or leverage, sales growth, capital intensity, cash flow | Non-linear U-shaped relationship | 2003–2015 | 7625 firm-years (1640 firms) |
| (Damert et al. 2017) | Carbon performance | Carbon intensity, Carbon exposure | Financial performance | ROA, ROE | Company size | Negative | 2008–2013 | 45 firms |
| (Nishitani et al. 2017) | Environmental performance | GHG emissions reduction, pollution emissions reduction | Financial performance | Profit growth, sales increase, productivity improvement. | Firm size, type of firm, market orientation, supply chain area, type of industry | Positive | 2009 | 100 firms |
| (Rokhmawati et al. 2017) | GHG emissions | Intensity of $CO_2$ | Financial performance | Return on Sales (ROS) | Firm size, leverage, capital intensity | Positive | 2010–2011 | 134 firms |
| (Capece et al. 2017) | Environmental performance | $CO_2$ emissions | Economic performance | ROI | | Positive | 2008–2013 | 237 firms, 1422 firm-years |

**Table 6.** *Cont.*

| Authors | Independent | | Dependent | | Controls | Correlations | Year Sample | Number of Samples |
|---|---|---|---|---|---|---|---|---|
| | **Variables** | **Indicators** | **Variables** | **Indicators** | | | | |
| (Rokhmawati and Gunardi 2017) | GHG emissions | Carbon intensity | Financial performance | ROE, ROI, ROS, Tobin's Q | Firm size, leverage, capital intensity, industry type | Positive | 2011 | 102 firms |
| (He et al. 2016) | Carbon performance | Carbon emissions/sales value, carbon disclosure index | Financial performance | Tobin's Q, ROA | Financing activities, leverage and firm size, capital intensity, sales growth, age of equipment, industry indicators, and year indicators. | Positive | 2007–2010 | 620 firms |

*4.4. The Motivations, Drivers, and Barriers to Corporate Carbon Strategies and their Outcome*

When analyzing corporate carbon strategies, it is not enough to solely study the practices and strategies implemented by companies. It is also crucial to understand the motivation, drivers, and obstacles they encounter during the transition. This understanding is essential for developing effective strategies for corporations and for governments to create policies that facilitate the adoption and implementation of emissions reduction initiatives by corporations. Based on the literature reviewed in this study, supplemented by additional literature (Schultz and Williamson 2005; Okereke 2007; Damert et al. 2017; Damert and Baumgartner 2018; Trumpp and Guenther 2017; Tuesta et al. 2020; Tahat and Mardini 2021; Busch et al. 2022), we have developed a framework that elucidates the relationship between theories, motivations, drivers, barriers to corporate carbon strategies, and their impact on financial performance, as illustrated in Figure 4.

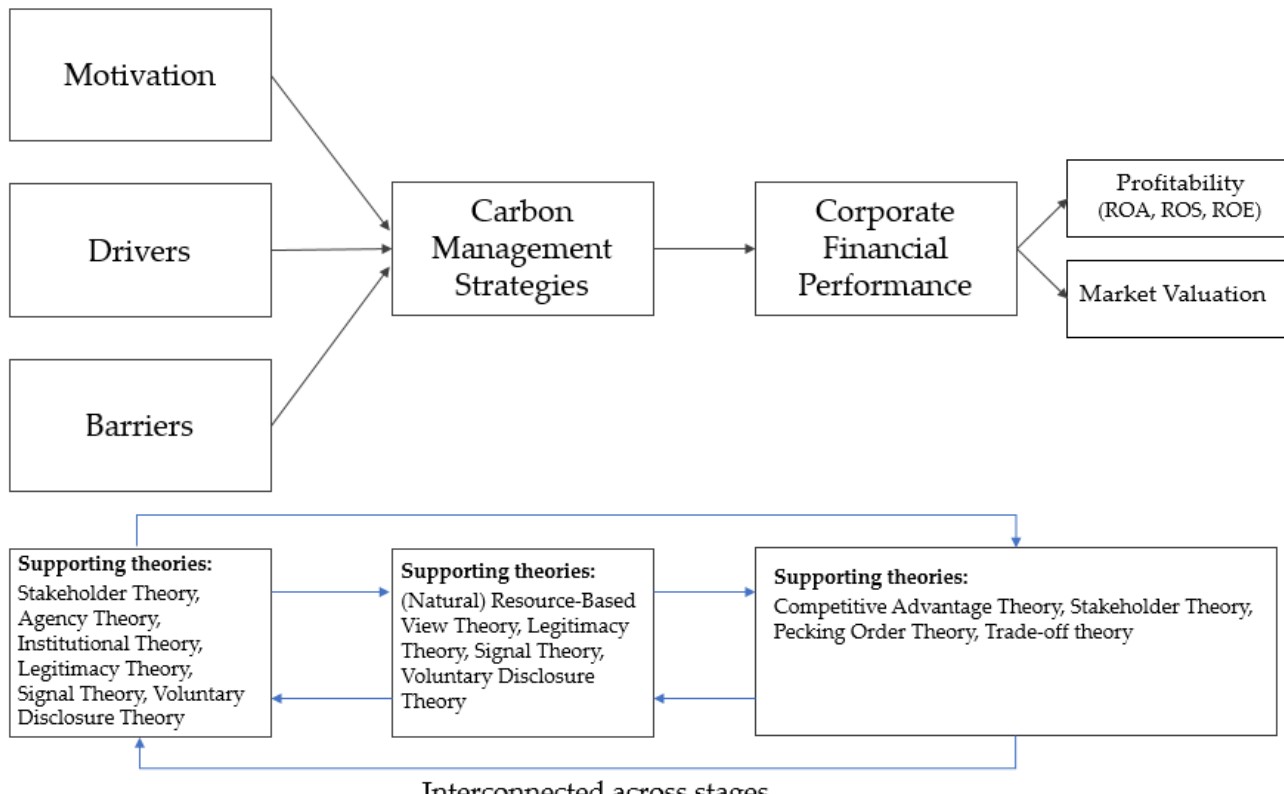

**Figure 4.** The framework delineates the relationship between theories, determinants (motivation, drivers, and barriers) of carbon management strategies, and their impact on financial performance.

In addition to the framework we have provided, we have developed an outline of the motivations, drivers, and barriers that corporations need to understand to successfully implement carbon management strategies in their companies. This summary is based on a review of relevant literature conducted in this research. Furthermore, we have included a summary of the current practices of carbon management strategies being implemented by companies, which can serve as a reference for those embarking on this journey. These summaries can be found in Table 7.

**Table 7.** The corporate motivation, drivers, barriers to carbon reduction strategies, and some examples of the strategies.

| Determinant Aspects | Detailed Practices |
|---|---|
| Motivation | Gain profitability <br> Reduce operational costs <br> Reduce business risks <br> Enhance reputation and brand image <br> Responsibility to mitigate the risk of climate change <br> Credibility to leverage in climate policy development <br> Ethical consideration <br> Developing market opportunities <br> Achieving competitive advantage |
| Drivers | Energy prices and costs <br> Market shifts <br> Regulation and government directions. <br> Investors' pressure <br> Financial institutions' pressures <br> Customer/consumers expectations <br> Technological shift and innovation <br> Business competition <br> Carbon Tax and financial penalties |
| Barriers | Lack of strong policy framework <br> Uncertainty about the government's action and regulation <br> Lack of law enforcement on compliance <br> Uncertainty about the marketplace <br> Short-term profit maximization <br> Lack of leadership commitment |
| Strategies | Corporate carbon policy and measurement <br> Product innovation <br> Process innovation <br> Logistics improvement <br> Energy efficiency initiatives <br> Switching to renewable energy <br> Planting trees and conservation <br> New market and product development <br> Participation in emissions trading scheme <br> Carbon compensation through carbon credits |

## 5. Conclusions

Based on our analysis of this SLR, we have observed that 59% of the articles demonstrate positive results. Among these, 50% show a significant positive impact, while 9% exhibit mixed results with both positive and negative outcomes in the short and long-term perspectives. These findings indicate that adopting carbon management strategies has a predominantly positive influence on corporate financial performance.

We believe that by expanding the scope of research to include a longer time period, increasing the sample size, and incorporating studies from beyond 2023, we can further strengthen the significance of carbon strategies. This is particularly relevant given the growing commitment of global corporations, countries/regulators, and global financial institutions towards achieving net zero emissions by 2050. The supportive policies and the increasing trend in this direction suggest that future SLR studies will likely yield more articles demonstrating a positive impact of carbon management strategies on corporate financial performance.

This SLR aims to enhance our understanding of the impact of carbon management strategies on financial performance. The findings of this study will provide valuable guidance to both practitioners and managers on how corporations can achieve a sustainable competitive advantage (Lewandowski 2017). To ensure a systematic approach, we adopted

the PRISMA methodology to structure our review and selection process. Following the four defined steps of the PRISMA framework, we identified and selected 22 articles for inclusion in our analysis.

*5.1. The Various Brief Results and Conclusions of the Study*

More than 50% of the articles reviewed in this analysis reported a positive and significant relationship between the implementation of carbon reduction strategies and financial performance. This finding can be attributed to the understanding that companies can benefit from costs savings in materials, energy, and services, as well as opportunities to increase revenues through products differentiation, better market access, and access to capital from financial institutions (Hoffman 2005; Derwall et al. 2005; Ambec and Lanoie 2008; Kumar 2018; Busch et al. 2022).

On the other hand, more than 27% of the articles indicated a negative relationship, suggesting that carbon management practices can impact profitability indicators such as return on assets (ROA) (Tuesta et al. 2020). This perception stems from the belief that "going green" comes with a cost. Achieving an ideal situation requires significant investment, time, and compliance with policies and regulations (Hang et al. 2019). Stricter regulations and evolving policies sometimes impose financial burdens on industries, particularly those in sensitive sectors, leading to high compliance costs (Qian et al. 2020). Additionally, investments in carbon competitiveness have been found to negatively influence financial performance (Damert et al. 2017).

The study (Lewandowski 2017) revealed that the link between carbon emissions and financial performance is positive for companies with strong carbon performance but negative for companies with weak carbon performance. This suggests that companies can reap financial benefits only after surpassing a certain threshold of carbon performance. Considering the increasing expectations from stakeholders regarding climate change efforts and the likelihood of even stricter regulations in the future, the focus of the debate should shift from discussing "whether or not" it pays to be green to "when" the corporations can offset the costs of environmental investments (Orsato 2006).

*5.2. The Future Research Opportunities and Research Limitations*

The European Union (EU) countries have demonstrated the most advanced practices and commitment to addressing climate change. This region has been at the forefront of implementing the EU Emissions Trading System (EU-ETS), which was the first and is, to date, the largest global emission trading system in the world (Brouwers et al. 2016). In contrast, countries such as Australia, Japan, South Korea, Norway, and Switzerland rely more on carbon fines as a regulatory mechanism (Tahat and Mardini 2021).

For future research opportunities, it is crucial to examine the impact of carbon strategies on the financial performance of corporations in developing countries, i.e., in the ASEAN region, where climate policy instruments are not yet widely established. In recent years, some developing countries have started introducing and implementing similar instruments and policies. For instance, Indonesia, Asia's emerging market, is introducing new regulations for the limited implementation of an Emissions Trading System (ETS) in the energy and power industries, with plans to expand its scope to other sectors beyond 2024 (ICAP 2023). The implementation of a carbon tax is also being considered (IMF 2022).

Future research should focus on exploring the effects of climate policies and regulations on companies' behavior, including how different aspects of carbon strategies are selected and prioritized to reduce carbon emissions. Understanding the impact of these strategies on companies' economic performance is of utmost importance. Additionally, further investigation is necessary to comprehend the motivations and commitment of multinational corporations operating, both at their headquarters and overseas branches. Examining whether these overseas branches exhibit an equivalent level of commitment to integrating corporate carbon strategies into their core business operations as their coun-

terparts at the headquarters is crucial, enabling comparisons between the developed and developing regions.

Furthermore, for other future research opportunities, it is necessary to examine whether the relationship between carbon strategies and financial performance differs between companies operating in advanced countries and those in developing countries and whether this relationship yields a positive or negative outcome. Finally, studying the impact of government-provided incentives on companies' efforts to adopt and implement carbon management strategies is crucial. Will these incentives motivate companies to accelerate their adoption of climate change strategy? Can these incentives directly or indirectly improve corporate financial performance? These potential research questions require further investigation.

We acknowledge several limitations that could potentially affect the outcomes of this SLR. Firstly, the use of keywords in this study was limited to journals within the Scopus database, which may have led to the exclusion of relevant articles indexed in other research databases. To enhance the comprehensiveness of future research, we recommend expanding the search to include additional publication sources, thereby deepening our understanding of the impact of carbon management strategies on financial performance in corporate settings.

Additionally, we believe that research on carbon strategies is expanding, particularly in relation to the use of the term "net zero" within the corporate context. Therefore, we suggest including the term "net zero emissions" as a keyword in future research for SLRs in the years beyond 2023. This is important because, in recent years, we have observed a growing popularity of this term among corporates when referring to their commitment in the context of corporate carbon strategies.

We anticipate that this article will enrich theory and knowledge regarding the relationship between carbon management strategies and financial performance in various industries or corporations. Furthermore, we aspire to have this paper included in a high-impact journal database such as Scopus.

**Author Contributions:** M.S., A.I.S., U.S. and N.Z. have contributed equally to this editorial. All authors have read and agreed to the published version of the manuscript.

**Funding:** This research received no external funding.

**Informed Consent Statement:** Not applicable.

**Data Availability Statement:** Not applicable.

**Conflicts of Interest:** The authors declare no conflict of interest.

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
