# Peer review of "Revisiting the Impact of Corporate Carbon Management Strategies on Corporate Financial Performance: A Systematic Literature Review"

_economies, doi:10.3390/economies11060171_

Round 1

Reviewer 1 Report

The aim of the research is fascinating and original.

The paper is well-structured and clear. Also the findings are very interesting.

I have only one curiosity and some suggestions:

I would like to know why you chose to analyse only the articles and not also the books and book chapters. Please explain your reasons in the paper.

Moreover, I suggest to highlight in the introduction and in the conclusion that this is the first paper on this type of analysis.

In the end, starting from your results, could you try to provide a framework that includes theory, determinants and companies' motivations, drivers, and barriers to corporate carbon management strategies?

The paper is well written

Reviewer 2 Report

The review of papers on offers an interesting insight into the still contested view of financial value.

Below I have highlighted specific comments relating to the paper text but there are a few more general points that I hope are useful:

1. Firstly, you have not used Net Zero as a search term within your Scopus review. Whilst this is relatively new in terms of corporate use it has become a major strategic approach. It would be useful to note why you have not included.

2. I wonder within your review you whether you should highlight that the systematic Literature Review is not able to ascertain the nature of ‘carbon management strategies’. Here I am considering the very varied approached that can be taken by companies under this descriptor– i.e. a focus on easy/cheap energy efficiency measures, small annual % reductions vs a full net zero carbon approach. The financial implications of these different approaches are considerable and will certainly impact negative or positive financial outcomes in short/long term. (consideration in Intro, section 4.2 and conclusion)

3. Please also note that when citing a writer(s) within the flow of copy you would state eg ‘Jones (2021) suggests’.  i.e. the name is in text, the date of publication is in brackets afterwards. You are not using this format in your paper and this needs to be corrected.

Specific Comments:

Ln 28 – weak opening sentence – please amend

Ln 29 – you state that anthropogenic emissions are ‘believed’ to be  - I would suggest that reports by the IPCC have moved beyond this point to confirm It is unequivocal that human influence has warmed the atmosphere, ocean and land’ (IPCC, 2021) – from Summary for Policymakers, Sixth Assessment Report, Climate Change 2021: The Physical Science Basis

https://www.ipcc.ch/report/ar6/wg1/resources/spm-headline-statements

Ln 30-34 – it would be helpful if you could reduce/amend this sentence – it is too long and looses the meaning you are trying to convey.

Ln 34-35 – the sentence needs to be clearer – not sure what it is referring to – reference required.

Ln 40 – would recommend not starting the sentence with ‘Nowadays’  - instead start with Increasing

Ln 42 – adopting carbon ‘emissions’ reductions or even GHG emissions reductions?

Ln 44 consider ending sentence after carbon management strategy.

Ln 48 – improve English – ‘are also known as’

Ln 50 – please ensure CO2 is written as CO2 throughout the paper

Ln 55 – consider starting sentence Nowadays with ‘An increasing number of’  or similar

Ln 56 – start of sentence needs to be improved

Ln 61 – suggest that ‘usual and have not yet transitioned to carbon reduction commitments.’

Ln 62 – would it be more correct to note that ‘key reason is identified as the pervasiveness’

Ln 66 – reference required for the believe by corporates that CRS will erode profitability

Ln 69 – suggest you start this sentence with a more positive position – ‘Our research indicates that there are’  or something similar

Ln 74 – you bring in ‘sustainability’ at this point. Are you enriching the theory across the whole topic of sustainability  - which is obviously much broader then carbon emissions. Are you looking at one aspect? – do you need to make this point clearer?

Ln 94  - suggest should read face rather than facing?

Ln 95  - their rather the carbon strategies?

Ln 108 – 112- sentence too long and no punctuation at end

Ln 112 – I would suggest you don’t need to identify in text who developed PRISMA, you make this clear by the reference.

Ln 124 – restrict rather than strict

Ln 125-126 – suggest be more succinct – ie Scopus has over 20,000 journals,  - remove information about database and academic studies from middle of sentence

Ln 128. Start of sentence may read better if is begins ‘Articles which are not full accessible are ……

Ln 128 – please make a very brief statement on why you are not listing articles that not fully accessible

Ln 128-129 – improve sentence structure

Ln 130  - recommend remove text ‘the very latest research studies’  - also assume this is ‘corporate’ commitments to climate change?

Ln 144- we suggest you do not need to repeat date

Ln 164- rather than we do the third step – perhaps we undertook

Ln 164-167 – sentence too long

Ln 175 – suggest sentence would read better if ‘ After this screening 34 articles remained’ ?

Ln 178 – suggest remove ‘to conclude the final selection of the articles reviewed’  and slightly rework section 2.4

Ln 189-192 – appears to repeat content already presented in the inclusion

Ln 217 – think you mean primarily rather than majority?

Ln218 – perhaps indicates rather than is showing the fact

Ln 228-229 – please improve sentence construction and provide a reference

Ln 236 – not a sentence  - please rewrite

Ln 237 – do you mean to say This carbon disclosure has become?

Ln 241 – assume you intend to state European Union countries?

Ln 286 – please could you check how many theories you are presenting – I feel it may be 5 rather than 4?

Ln 358 – please amend presentation of table – will the journal allow this to presented in landscape? Please ensure headings appear on each page. Also it would be helpful to think about the best way I which to group papers – here you have presented by date, but could correlation be more useful as the focus of your review is on financial impact? Perhaps then within correlations by another variable or indicator you feel is important? You may want to highlight your logic at the start of the table so the reader understands the material more clearly.

Ln 372 – we would suggest that ‘develop our understanding on’  - perhaps we develop an outline of’?

Ln 383 – I don’t believe you have used ‘environmental strategy’ previously – do you want to bring in a new term at this point?

Ln 392-393 – should the point about methods, timeframes etc be a key point in your earlier discussion of findings?

5.1 – general point that lack of standardisation in targets and reporting make comparisons difficult

5.2 – not including net zero may be a limitation?

1. The standard of written English is generally good but I would recommend a re-read of all text and amendment especially to, the Introduction and Methodology sections.

2. Please make sure you continue to write in the same tense throughout  - you do move between present and past tense incorrectly in places

Reviewer 3 Report

Well, the overall draft of the paper is a reasonable effort by the authors, but I have some major suggestions:

1. The introduction section should be re-written/revised carefully i.e., it is suggested that in the introdution section focus more on the significance of the study with strong arguments and main contributions of the study toward literature in a concise manner.

2. I think Scopus database is not enough for such study so i suggested that WOS should be included.

3. In the conclusion section, how practically the firms, investors and policy makers used the findings and also the explain the future direction for the scholars should be addressed.

4. Lastly, literature section missing article from 2023, so the authors may take help from these latest papers or search by theirself.

https://doi.org/10.1111/basr.12301

https://doi.org/10.1016/j.ribaf.2023.101974

https://doi.org/10.1111/acfi.13090

Can be improved

Reviewer 4 Report

1. you can explain your filtering the data till you choose 22 article for your research

2. In introduction you need explain more about this topic and the development of this topic right now so make a literature review on this topic is important

3. The passage provides a good introduction to the topic, explaining the relevance of climate change and the role corporations play in it. However, the narrative is a bit dense, with many references. Try to simplify the narrative while retaining the main points.

4. I suggest slightly rephrasing RQ5 to make it clearer. For instance, you could divide it into two parts: "What are the opportunities for future research to advance academic debate on corporate carbon management?" and "What are the limitations of existing literature on this topic?"

5. The section explaining the organization of the article could be simplified for easier understanding. For instance, "This article begins with an explanation of our research methodology, following the PRISMA guidelines for systematic literature reviews (Section 2). The results and discussion of the study are presented in Section 3, including answers to our research questions. Section 4 concludes the paper, outlining future research opportunities and acknowledging limitations of our research."

 6. Remember, keeping your narrative clear and simple helps your readers to better understand your research. 

7. The English in the document is generally good but there are some instances of awkward phrasing and unclear sentences, such as in line 166-167 ("From checking the title, we exclude some articles that are not meeting the scope of the objectives studied."). It would be helpful to make these clearer and more concise.

8. The PRISMA flowchart illustrating the process of article selection would be beneficial to visually represent the process of inclusion and exclusion of articles for the review.

9. The text could benefit from a bit more contextualizing information about the Paris Agreement, as it is central to your study. For example, briefly explaining its purpose and the commitments it calls for would be helpful for the reader.

10. consider more explicit signposting throughout the paper. Although your paper has a clear structure, the use of headings and sub-headings will greatly improve the readability and flow of the paper. For instance, consider introducing sub-headings for the different parts of your methods section (Identification, Screening, Eligibility, Inclusion and Exclusion). 

11. In some places, the sentences are quite long and could be broken up for clarity. For example: "The questions that we want to understand and discuss are the theories used in this carbon strategies and financial performance relation, the result study of the recent related literature, the common variables and indicators used in the recent studies, and understanding the motivations, drivers, and barriers that the companies experience in implementing the corporate carbon strategies." This sentence contains multiple ideas and could be made more reader-friendly by breaking it down into smaller sentences or bulleted points.

12. In places, the language is quite passive, which can lead to ambiguity. For instance, "The last one is the use of the (natural) resource-based view theory that suggested the firm use all its resources, investments, differentiation, and uniqueness to yield competitive advantages." Here, it is unclear who or what is using the resource-based view theory. More active language could clarify this.

13. It would be beneficial to add more transitional phrases and signposting to guide the reader through your arguments. This helps to ensure that the logic of your arguments is easy to follow.

14. The summary of the articles' findings about the relationship between corporate carbon management and corporate performance could be clearer. Some of the phrases used are a bit vague, such as "there is also the result that depends on the time perspective." More specific language would help the reader understand the results of these articles better.

14. It would be helpful if "Figure 3" show more details for the readers to visually comprehend the distribution of result study on the relationship between Carbon Performance and Financial Performance.

need check grammar and proofread

Round 2

Reviewer 3 Report

The authors added the required suggestions carefully so i recommend for publication, conglts authors.

Still need to improve English

Reviewer 4 Report

all my comment already addresed and this paper suitable for publish in economies

minor modified